# Soil Component: A Potential Factor Affecting the Occurrence and Spread of Antibiotic Resistance Genes

**DOI:** 10.3390/antibiotics12020333

**Published:** 2023-02-04

**Authors:** Hongyu Shi, Xinyi Hu, Wenxuan Li, Jin Zhang, Baolan Hu, Liping Lou

**Affiliations:** 1Department of Environmental Engineering, Zhejiang University, Hangzhou 310029, China; 2Key Laboratory of Water Pollution Control and Environmental Safety of Zhejiang Province, Hangzhou 310020, China

**Keywords:** soil component, antibiotic resistance genes, horizontal gene transfer

## Abstract

In recent years, antibiotic resistance genes (ARGs) and antibiotic-resistant bacteria (ARB) in soil have become research hotspots in the fields of public health and environmental ecosystems, but the effects of soil types and soil components on the occurrence and spread of ARGs still lack systematic sorting and in-depth research. Firstly, investigational information about ARB and ARGs contamination of soil was described. Then, existing laboratory studies about the influence of the soil component on ARGs were summarized in the following aspects: the influence of soil types on the occurrence of ARGs during natural or human activities and the control of exogenously added soil components on ARGs from the macro perspectives, the effects of soil components on the HGT of ARGs in a pure bacterial system from the micro perspectives. Following that, the similarities in pathways by which soil components affect HGT were identified, and the potential mechanisms were discussed from the perspectives of intracellular responses, plasmid activity, quorum sensing, etc. In the future, related research on multi-component systems, multi-omics methods, and microbial communities should be carried out in order to further our understanding of the occurrence and spread of ARGs in soil.

## 1. Introduction

Antibiotics are widely used and have low bioavailability, leading to their continuous release into the environment [1,2]. The soil environment is an important acceptor of many pollutants, including antibiotics. The external pressure of antibiotics and other pollutants can promote the proliferation and occurrence of antibiotic-resistant bacteria (ARB) and antibiotic-resistance genes (ARGs) in soil [3]. Soil ARGs can spread in various ways, such as horizontal gene transfer (HGT) between soil microorganisms and vertical gene transfer (VGT) between parent and offspring [4].

Different soil types and soil components could cause significant differences in the occurrence and propagation of ARGs in soil. Some researchers have found that the influence of human activities on the occurrence of ARGs in soil is related to soil type [5,6], and the influence of soil type and related properties is even greater than that of human activities themselves [7,8,9], while others have achieved the control of ARGs in soil by the exogenous addition of soil components such as biochar [10] and natural zeolite [11]. Studies from the micro perspectives discussed the influence of soil components on the horizontal gene transfer process of ARGs including transformation [12] and conjugation [13] in pure bacteria systems based on the pure culture of bacteria. Relevant studies have been carried out from the above perspectives, so we could deepen our understanding of soil types and soil components affecting the occurrence and spread of ARGs and lay a foundation for future research by collating and reviewing these contents.

Based on an overview of ARGs’ occurrence and spread in soil, this paper focused on the effects of soil type and soil components on the occurrence of ARGs and deeply analyzed the mechanisms of soil components on HGT progress, which is expected to provide a precious reference for in-depth research on soil ARG pollution control and the reduction of its ecological risk as soon as possible.

## 2. The Occurrence and Spread of ARGs in Soil

### 2.1. The Pollution Status of ARB and ARGs in Soil

Soil is the largest reservoir of ARBs and ARGs [14,15]. The abundance of ARGs in soil has increased substantially since the beginning of the antibiotic era [16,17]. Similar to antibiotics, wastewater irrigation and manure application are two main routes for ARBs and ARGs entering the soil [15,18].

ARBs and ARGs have been widely detected in different types of soils around the world [19]. Taking *E. coli*, which has been widely studied, as an example, the existence of *E. coli* has been found in farmland soil, non-farmland soil, and even plant microbial communities [20,21]. Many *E. coli* strains isolated from soil carry ARGs, and most of them have multidrug resistance. Furlan et al. [22] isolated a total of 60 strains of *E. coli* from soil samples on Brazilian farms, of which 68.3% of them exhibited multidrug resistance profiles. Liu et al. [21] found that all soil *E. coli* isolated from Washington State (*n* = 1905) were resistant to at least four different antibiotics. Graves et al. [23] analyzed 616 strains of *E. coli* collected from swine manure, swine lagoon effluent, and soils that received lagoon effluent and found that these strains usually carried ARGs coding for streptomycin, spectinomycin, tetracycline, and sulfonamide.

At the same time, the relative abundance of various ARGs in soil has increased significantly in recent years, and the increase in tetracycline resistance gene levels was significantly more frequent than for other ARGs [24]. Tetracyclines ARGs are present in soils worldwide, and the relative abundance is between 10^−9^ and 10^−2^ copies/16S rRNA gene (Table 1). Tetracyclines ARGs are also the main types of ARGs carried by phages in the soil environment [25].

### 2.2. Transmission Routes of ARGs

The causes, inheritance, and transmission mechanisms of antibiotic resistance are very complex and can be divided into genetic resistance and non-inherited resistance [39] (Figure 1). Non-inherited resistance refers to resistance that is not acquired through horizontal or vertical transfer of ARGs but through behaviors such as collaboration between groups [40]. Cooperative resistance, a population-based survival strategy that adapts to high antibiotic stress through the cooperation of multiple ARBs, is a typical non-inherited resistance [41]. Cooperative resistance widely occurs in infections of the upper respiratory tract, skin, and soft tissue, resulting in many cases of antibiotic treatment failures and polymicrobial infections, which have attracted a lot of attention in clinical studies [42]. However, cooperative resistance in the natural environment has not received enough attention, so we summarized and discussed the spread of antibiotic resistance from the perspective of genetic resistance.

Genetic resistance can be divided into intrinsic resistance and acquired resistance. Intrinsic resistance refers to the natural existence of certain genes in bacterial genomes that could generate a resistance phenotype [43]. It is an ancient, natural, and widespread environmental phenomenon that predates the selective pressures caused by modern human use of antibiotics, and multiple ARGs have been detected in Arctic permafrost unaffected by human activities [19,44]. Acquired resistance is a consequence of spontaneous chromosomal mutations or ARGs gained through HGT [45], which refers to the exchange of genetic material between individuals of different organisms and is the key reason for the widespread existence of ARB in clinical systems [46]. After obtaining ARGs through HGT, ARGs will achieve the amplification and continuation of these genes through reproduction between parent and child generations in VGT [47].

HGT mainly includes three pathways mediated by mobile genetic elements (MGEs), namely extracellular DNA-mediated transformation, plasmid-mediated conjugation, and phage-mediated transduction [46]. A large number of research results have shown that HGT can widely occur in the soil environment [48,49,50].

Transformation refers to the process by which competent bacteria take up DNA from outside. Unlike conjugation, transformation does not require physical contact between the donor and recipient cells, and free DNA released by cell lysis can serve as the donor for transformation [51]. Only competent bacteria can obtain extracellular DNA, and the competence can be naturally or artificially induced [52]. Most naturally transformable bacteria can develop into competent cells under specific circumstances (e.g., nutrient conditions, changes in bacterial density [53]). Johann et al. [54] listed 87 species of bacteria that can absorb extracellular free DNA through natural transformation, including *Pseudomonas* and *Acinetobacter,* which are commonly found in soil. The key steps of ARGs transformation are as follows: (1) bacteria actively or passively discharge ARGs into the environment; (2) extracellular ARGs become stable and ingestible in the environment; (3) extracellular ARGs are ingested into the bacterial cytoplasm; (4) exogenous ARGs integrate into bacterial chromosomes by homologous recombination or replicate autonomously as episomes [51,52,53,55]. Chen et al. [56] found that the plasmid pK5 carrying ARGs had a strong migration ability in soil, which confirmed the widespread occurrence of the transformation process in soil.

Conjugation refers to the process by which the plasmid or chromosome carrying ARGs enters the recipient bacteria through the conjugative fimbriae produced by the donor bacteria [57,58]. Conjugation is considered to provide better protection from the surrounding environment and a more efficient means of genetic material entering the host cell than transformation, while often having a broader host range than bacteriophage transduction [46]. Integrative and conjugative elements (ICEs) and plasmids are the main vectors for the delivery of ARGs in conjugation [59,60,61]. Plasmids, as important mediators of conjugation, can still persist among bacterial populations without antibiotic stress and invade new strains with high frequency [62,63]. It can be classified into three categories according to mobility: conjugative, mobilizable, and non-mobilizable [59]. A conjugative plasmid codes for its own set of mating pair formation (MPF) genes; if it uses an MPF of another genetic element present in the cell, it is called mobilizable; other plasmids are called non-mobilizable because they are neither conjugative nor mobilizable and usually spread through transformation and transduction [59,61]. In addition, non-mobilizable plasmids can also be transferred by physical association with conjugative plasmids [64]. Plasmid-mediated conjugation includes multiple processes such as mating pair formation and relaxosome formation [61,65]. In soil, this process is susceptible to a variety of factors, such as soil bacterial population structure [66], nutrient composition [67], selective pressure of antibiotics and heavy metals [68], etc. However, the understanding of the plasmid-mediated conjugative transfer process of the complex bacterial community in soil is still limited [69]. ICEs have the properties of transposons, plasmids, and phages: both ICEs and transposons can jump on chromosomes, but transposons cannot undergo HGT; both ICEs and plasmids can transfer DNA between cells in the form of conjugation, but most ICEs cannot self-replicate as plasmids do; ICEs and phages both can detach, integrate, and replicate with host chromosomes, the difference being that ICEs transfer DNA in the form of conjugation rather than transduction [70]. The study by Gonçalves et al. [71] confirmed the role of ICEs in the soil microbial HGT process.

In transduction, ARGs are transferred from one bacterium to another by means of phages (bacteriovirus) and can be integrated into the chromosome of the recipient cell [72]. The phage-mediated transduction progress does not need contact between the donor and recipient, or even the simultaneous appearance of them [73]. Soil is one of the important habitats for phages and their hosts [74,75]. It is estimated that the number of soil virus particles (mainly phages) accounts for 10% of the total number of viruses in the world, about 4.8 × 10^30^ [76]. The special protein capsid structure of phages can effectively protect nucleic acids, and the soil is highly heterogeneous and rich in biodiversity, providing a variety of parasitic environments for phages [50]. The opaque environment of soil protects the phages from sun damage [77], which is more conducive to their long-term survival and reproduction. Therefore, compared with free ARGs and bacteria in soil, phages are more resistant to adverse environmental factors [78] and persist longer, providing a material basis for their interactions with bacteria and gene transfer. Related studies have shown that the contribution of phages to HGT was likely underestimated [79,80]. Olatz et al. [81] found a large number of free and replicable phages containing ARGs in farmland soil, which may lead to the production and enrichment of ARB.

Although HGT is regarded as the major pathway of ARGs spreading, there is a significant involvement of VGT. Firstly, VGT raises the possibility of spontaneous mutation of bacterial DNA [55]. Secondly, VGT promotes HGT among the bacterial community: Li et al. [4] found that VGT can significantly promote the formation of conjugants and accelerate the spread of ARGs.

## 3. Effects of Soil Types and Soil Components on the Occurrence of ARGs from the Macro Perspective

Existing studies about the effects of soil types and soil components on the occurrence of ARGs and HGT mainly include the following perspectives: Explore the effects of soil types on the occurrence of ARGs and the control of exogenously added soil components on ARGs from the macro perspective, or focus on the effects of soil components on the HGT of ARGs in a pure bacterial system from the micro perspective. In this section, we first discussed the effects of soil types and soil components on the occurrence of ARGs from the macro perspective.

### 3.1. The Effects of Soil Types on the Occurrence of ARGs

Soil types have important effects on ARGs abundance, composition, and distribution (Table 2). The influence of soil type and its own properties on the abundance of ARGs even exceeds that of human activities such as long-term grazing [7], wastewater irrigation [8], and composting [9]. Although Feng et al. [82] found that soil types were not as influential as corpse decomposition, environmental factors such as NH_4_^+^ concentration and pH were still the main reasons affecting ARGs, and these environmental factors were all related to soil types. When human activities have an impact on the abundance of ARGs, the degree of impact also varies with soil types. For example, Wang et al. [5] found that higher diversity and abundance of ARGs occurred in fluvo-aquic and saline-alkali soil than in cinnamon soil after long-term manure application; Zhang et al. [6] also found that the enrichment of ARGs in long-term manured soil was influenced by pH. The effect of soil types on ARGs is mainly through the following pathways: Soil types significantly influence the soil properties, which change the composition of the microbial community and ultimately reflect in the abundance and species of ARGs; some studies have also suggested that soil types directly affect microbial communities [83], thus changing ARGs. The soil environment is complex, and the influencing factors are diverse, so it is difficult to clarify the specific impact path of ARGs, but it is clear that soil types have a notable role in ARGs pollution and should be paid attention to. The soil type is closely related to the content of each component in the soil, and the study of a single component is helpful to explain the mechanism.

### 3.2. The Control of Exogenously Added Soil Components on ARGs

At present, there are few related studies on the effect of a single soil component on ARGs, and some of them have used exogenous addition methods to explore the effects of biochar, natural zeolite, and other components on the spread of ARGs (Table 3). Biochar is one of the main sources of soil black carbon [84], and zeolite is a widely distributed silicate mineral [85]; both of these are common soil components. Researchers have found that biochar has a certain effect on the macro-control of ARGs pollution and migration [10,86,87,88,89,90], which is mainly reflected in hindering or even blocking the horizontal or vertical migration of ARGs in soil, improving soil properties and structure, and reducing the selection pressure of heavy metals and antibiotics [10,11,89,91]. However, some studies also pointed out that the control effect of biochar is limited [92]. Studies on natural zeolite have drawn similar discrepancies [11,86]. The purpose of these studies is mainly to control ARGs in soil, while soil components such as biochar and zeolite are considered widely used and environmentally friendly options. In addition to single-component addition, it is also worth exploring the effects of multi-component mixed application on soil ARGs and whether it can achieve more effective soil ARGs control.

**Table 2 antibiotics-12-00333-t002:** Research on the effects of soil types on antibiotic resistance genes (ARGs) abundance.

Soil Types	Research Process	Important Conclusions
Red, yellow, and black soils	Corpse decomposition [82]	Soil types have few impacts on ARGs; treatment, microbiome, NH_4_^+^ concentrates and pH are primary determinants of ARGs.
Loamy-sand, loam, and clay	Wastewater irrigation [8]	Soil type was the key factor in ARGs distribution; soil ARGs relative abundances were independent of the irrigation water quality
Fluvo-aquic, saline-alkali, and cinnamon soils	Long-term manure application [5]	Soil types influenced the ARGs distribution; higher diversity and abundance of ARGs occurred in fluvo-aquic and saline-alkali soil than cinnamon soil; Sand, pH and Zn contributed more to the pattern of ARGs in the cinnamon soils; sand and Cd, clay and Pb contributed the most in the fluvo-aquic and saline-alkali soils, respectively.
Acidic, near-neutral, and alkaline soils	Long-term manure application [6]	Soil types indirectly affected ARGs, while bacterial abundance and mobile genetic elements directly impacted ARG profiles; the effect of manure fertilization on the ARG profile in acidic and near-neutral soils was stronger than that in alkaline soil.
Humic acrisol, calcaric cambisols, and histosols	Interval fertilization [93]	Soil types affected ARGs.
Red soil, loess, and black soil	Fertilization [9]	The main contributor to the evolution of ARGs varied from soil types; no significant difference of antibiotic resistant bacteria and ARGs was observed among compost types.
Grassland soils	Long-term grazing [7]	ARGs shaped by the initial plant, soil environmental parameters (NO_3_^−^-N, TN, TP, pH) and microbiomes in grassland; long-term historic grazing had no effect on ARGs in grassland soils.
Sediments	Interannual variation [94]	TOC and clay were the major environmental factors regulating the variations in ARGs in sediments

**Table 3 antibiotics-12-00333-t003:** Research on the effects of externally adding soil components to antibiotic resistance genes (ARGs).

Environmental Medium	Exogenously Added Soil Components	Important Conclusions	
		Results	Reasons
Soil [87]	Biochar and pyroligneous acid	Both single and combined application of pyroligneous acid and biochar reduced the absolute abundance of ARGs in the rhizosphere and non-rhizosphere soils of leafy vegetables.	Pyroligneous acid and biochar reduced the bioavailability of heavy metal and improved soil properties.
Soil [10]	Biochar	Biochar impeded the vertical transport of ARGs.	Biochar addition enhanced dissolved organic matter export from soil, changed its composition.
Soil [92]	Biochar	Biochar amendment significantly decreased the abundance of ARGs in non-planted soil, but was not sufficient enough to alleviate ARGs level in planted-soil and plants.	Biochar was not sufficient enough to alleviate ARGs level; Increasing soil microbial diversity is more useful in mitigating ARG spread and accumulation.
Soil [91]	Biochar	*Lolium multiflorum* exhibited significantly stronger abatement of ARGs when combined with biochar than used alone; Soil pH and trace elements exerted weaker effects on ARGs after the application of biochar.	*Lolium multiflorum* and biochar improved soil physical structure, directly promoted the abatement of antibiotics and ARGs.
Soil and lettuce [88]	Biochar	Biochar reduced the relative abundance of ARGs in lettuce leaves, roots and soil.	The increased adsorption due to biochar and microbial degradation significantly alleviated environmental pressure; Bacteria were adsorbed, thus hindering their transport.
Soil and lettuce [89]	Biochar	Biochar can prevent soil antibiotics from accumulating in lettuce tissues; The enrichment of antibiotic resistant bacteria and the abundance of ARGs in lettuce was reduced by biochar treatment.	
Anaerobic digestion of swine manure [90]	Biochar	Biochar contributed to ARGs removal.	Biochar indirectly affected ARGs by changing *intI*1 and microbial structure.
Sludge composting [11]	Natural zeolite	Natural zeolite only controlled over some ARGs and had limited effect on bacterial community changes.	Porous structure of natural zeolite hindered microbial exposure and reduced heavy metal selection pressure.
Chicken manure composting [86]	Zeolite and biochar	Biochar and zeolite reduced the relative abundance of ARGs.	Biochar and zeolite had a suppressing effect on the abundance of *intI1*, and a reducing effect of horizontal gene transfer through conjugation and transformation.

## 4. Effects of Soil Components on the HGT of ARGs from the Micro Perspective

### 4.1. The Effects of Soil Components on the HGT of ARGs in Pure Bacterial System

Another part of the studies discussed the effect of soil components on HGT processes such as ARGs transformation and conjugation using the pure bacterial system (Table 4 and Table 5).

Existing studies on transformation not only discuss the changes of ARGs vectors (plasmids, chromosomes, etc.) and recipient bacteria in soil microcosms, EPS, sediments, or other media but also include studies on the addition of single components to simulate soil conditions [95,96,97,101]. Relevant research (Table 4) has shown that plasmids [12] and chromosomes [98] adsorbed by soil components can still participate in transformation; Chamier et al. [99] found that the plasmid adsorbed on sand transformed significantly less efficiently than the plasmid in solution; but Dong et al. [101] considered that sediment-adsorbed plasmids had higher transformation efficiency than episomal plasmids. Montmorillonite at low concentrations (0–0.025 g/L) [96] and goethite at high concentrations (10 g/L) [95] promote transformation, while high concentrations of kaolinite (10 g/L), montmorillonite (0.025–2 g/L and 10 g/L), and biochar (2, 4, and 8 g/L) inhibit it [95,96,97].

The research on conjugation (Table 5) showed that birnessite and low concentrations of goethite (0–0.5 g/L) promoted conjugation; the effects of kaolinite and montmorillonite were irregular; goethite at high concentration (5 g/L) inhibited conjugative transfer [13]. Liu et al. [102] found that biochar can weaken the promoting effect of heavy metals on conjugation, while Zheng et al. [103] reported that pyroligneous acid and its three fractions at different temperatures had inhibitory effects on conjugative transfer. Some studies illustrated the mechanisms of soil components affecting the process of conjugation by detecting the expression of related genes [13,102,104], but most of them are speculation based on transcriptome results, and the understanding of related pathways and mechanisms is still unclear, which is worth exploring in depth.

**Table 5 antibiotics-12-00333-t005:** Research on the effects of soil components on conjugation of antibiotic resistance genes (ARGs).

Medium	Results	Reasons
Kaolinite, goethite, birnessite, and montmorillonite [13]	Birnessite promoted conjugation. The effects of kaolinite and montmorillonite were irregular. Goethite promoted conjugation at low concentration (0–0.5 g/L) and inhibited it at high concentration (5 g/L).	Birnessite promoted the production of intracellular reactive oxygen species (ROS); increased the expression levels of oxidative stress-regulated genes (*rpo*S) and outer membrane protein genes (*omp*A, *omp*F, *omp*C). Birnessite altered the expression levels of conjugation-related genes (globally regulation genes (*kor*A, *kor*B, *trb*A); mating pair formation (MPF) system genes (*trb*Bp, *tra*F); DNA transfer and replication (DTR) system genes (*trf*Ap, *tra*J)).
Dissolved biochar [105]	The effects on conjugation were related to the concentration and source of biochar.	Humic acid-like substance in dissolved biochar improved the conjugative efficiency. The inhibitory effects of small-molecule matters dominated, decreasing conjugative transfer frequency.
Pyroligneous acid and its three fractions [103]	Reduced the abundance of ARGs and MGEs in soil.	High content of organic acids inhibited the bacterial growth.
Dissolved biochar [102]	Attenuated the promotion effect of Cu (Ⅱ) to conjugation.	Dissolved biochar affected intracellular ROS production level, cell membrane permeability, and the expression level of global regulatory genes (*kor*A, *kor*B, *trb*A), pore formation and membrane trafficking genes (*omp*A, *omp*C), MPF system gene (*trb*B), DTR system gene (*trf*A), etc.
CeO2 nanoparticle [104] (soil pollutant)	Inhibited conjugation at low concentration (1, 5 mg/L), while promoted it at high concentration (25, 50 mg/L).	CeO_2_ nanoparticle affected many aspects, such as intracellular ROS production, polysaccharide synthesis in EPS, cell-to-cell contact, ATP supply, and the expression level of conjugation-related genes (MPF system gene (*trb*Bp), DTR system gene (*trf*Ap), putative transmembrane ATPase gene (*tra*G)), etc.
Gut of *C. elegans* [106] (soil animal)	The conjugation efficiency in gut was higher than soil, and increased with time and temperature.	The abundance of MPF system gene (*trb*Bp) and DTR system gene (*trf*Ap) was increased.

### 4.2. Influence Mechanisms of Soil Components on HGT of ARGs

Although transformation, conjugation, and transduction are three independent HGT mechanisms, there are some commonalities between them when soil components are present. Soil components mostly affect the HGT process of ARGs through similar pathways: from the perspective of intracellular changes and responses, including regulation of intracellular reactive oxygen species (ROS) production, SOS response, and the expression levels of related genes, etc. [13,104,107]; from the point of view of intercellular contact and communication, it includes the influence of extracellular polymeric substances (EPS) [104] and quorum sensing [108,109], etc.; in addition, it also includes affecting the activity of plasmids or bacterial concentration [110,111,112].

#### 4.2.1. Intracellular Changes and Responses

##### Intracellular ROS Production

ROS are generated via successive single-electron reductions, including superoxide (O_2_·^−^), hydrogen peroxide (H_2_O_2_), and hydroxyl radical (OH·) [113]. Intracellular ROS generation can cause oxidative stress, which affects a series of macromolecules of bacteria (DNA, lipids, and proteins) [114]. Intracellular ROS can be scavenged by the antioxidant system, which is an intracellular defense mechanism [115]. Antioxidant enzymes (such as catalase (CAT) and superoxide dismutase (SOD)) catalyze the conversion and detoxification of corresponding oxidative groups and, finally, relieve oxidative stress [116]. Moderately generated ROS after treatment with soil components may stimulate a series of protective responses that favor the promotion of HGT. Birnessite can initiate the formation of intracellular ROS and induce oxidative stress, which is one of the important mechanisms for birnessite-promoting ARGs conjugation [13]. However, excessive production of intracellular ROS will exceed the capacity of antioxidant enzymes, resulting in severe cellular damage or death of cells, ultimately inhibiting conjugation [117].

##### SOS Response

SOS response is a global regulatory response to protect cells from severe DNA damage by ROS [118], which has been shown to promote the HGT of ARGs [107]. However, there are few studies on the induction of bacterial SOS responses by soil. It is speculated that the natural components in soil have limited influence on the bacteria, while the nanoscale components or other pollutants in soil may cause the excessive accumulation of ROS and induce the SOS response. For example, high concentration of nano-CeO_2_ (50 mg/L) caused the up-regulation of both SOS response activation genes (*lex*A, *rec*A) and DNA repair genes (*umu*C, *umu*D, *uvr*A, *uvr*B) [104], which promoted the conjugative transfer of ARGs.

##### Cell Membrane Permeability

Cell membrane permeability changes with the stimulation of environmental stress, and such changes are potentially related to the spread of genetic materials [97]. The increase in cell membrane permeability, which can be divided into active improvement and passive damage, may contribute to the transfer of ARGs to a certain extent [53,102].

On the one hand, under the action of soil components, bacteria can autonomously up-regulate the related gene expression of membrane proteins, that is, active improvement. For example, Wu et al. [13] found that birnessite up-regulated the expression level of several outer membrane protein genes (*omp*A, *omp*F, *omp*C), thus promoting the conjugative transfer of ARGs. On the other hand, bacteria may be physically damaged by external perturbations, resulting in the formation of pores on the cell membrane (e.g., collisions with bacteria during material mixing [119,120]); it is also possible that some soil components, especially nanoscale soil components (e.g., high-temperature black carbon), allow the excessive production of intracellular ROS and then damage the integrity of cell membranes; another possibility shows that the high concentration of heavy metals released from the process of interaction between soil components and bacteria indirectly promotes lipid peroxidation and induces cell membrane damage [121,122]; all of the above are passive damage. Both goethite [95] and montmorillonite [96] were found to promote the transformation of ARGs by causing cell membrane damage.

When the integrity of the bacterial cell membrane is excessively damaged, the bacteria will die, which inhibits the conjugation of ARGs. But the ARGs released from damaged or dead bacteria are free from the soil and have the opportunity to become donors of transformation. Ma et al. [123] and Ouyang et al. [124] reported that soil minerals, such as kaolinite, goethite, and hematite, can induce bacterial death by disrupting cell membranes. In addition, bacteria can also initiate protective responses by reducing cell membrane permeability, thereby reducing the uptake of toxic substances [125,126], while also hindering the occurrence of HGT. For example, biochar dissolutions caused a decrease in cell membrane permeability, thus inhibiting the transformation of ARGs [97].

##### ATP Synthesis Capacity

The construction of conjugative transfer apparatus, replication of plasmids, and transport across cell membranes all depend on adenosine triphosphate (ATP) [127]. Soil components can affect the frequency of conjugation and transformation by regulating ATP synthesis. For example, CeO_2_ caused an insufficient ATP supply, which in turn inhibited the process of conjugation of ARGs [104].

##### Conjugation Activity of Intracellular Plasmids

The conjugation of plasmids requires the participation of a series of conjugation-related genes and regulatory genes, such as global regulatory genes (*kor*A, *kor*B, etc.), DNA transfer and replication (DTR) system genes (*trf*Ap, etc.), and MPF system genes (*trb*Bp, etc.) [106]. Among them, the MPF system is crucial for the formation of fimbriae [128]. In Gram-negative bacteria, sexual fimbriae act as channels for DNA conjugative transfer, and both their length and flexibility affect the efficiency of bacterial contact, including collision, attachment, and detachment [129]. As to transformation, the adherence of high concentrations of mineral particles to bacteria may damage fimbriae, while its absence will greatly reduce the expression of competent genes and the formation of competent bacteria, thus affecting the transformation process of ARGs [95,130].

#### 4.2.2. Cell-Cell Contact and Quorum Sensing

The EPS consists of exopolysaccharides, nucleic acids, proteins, lipids, and other biomolecules, which determine the surface properties of bacteria (e.g., surface charge) and are critical for inter-bacterial adhesion and communication [131]. It was concluded by Tsuneda et al. [132] that, if the EPS amount is relatively small, cell adhesion onto solid surfaces is inhibited by electrostatic interaction, and cell adhesion is enhanced by polymeric interaction when it is relatively large. This process may affect the contact behavior between bacteria (e.g., conjugation) and may also enrich the plasmid or block the contact between the plasmid and bacteria, thereby affecting transformation. However, few studies have paid attention to the effects of soil components on bacterial EPS production and the transfer of ARGs, while relatively many studies have focused on soil pollutants. For example, Yu et al. [104] found that CeO_2_, a typical nanoparticle pollutant in soil, weakened inter-bacterial contact by inhibiting the synthesis of polysaccharides in EPS.

EPS can also act as a permeability barrier to limit the increase in cell membrane permeability and hinder the transformation of ARGs [133]. Wang et al. [100] found that the transformation ability of free ARGs was higher than that of activated sludge EPS when calculated by per ng DNA, and lower when calculated by per g volatile suspended solids (VSS). This phenomenon proved that although activated sludge EPS had a certain inhibitory effect on gene transfer. Due to the large amount of ARGs contained in EPS, it has a significant enrichment effect on ARGs, and may be an important environmental source of extracellular ARGs for bacteria.

Bacterial quorum sensing is a form of bacterial cell-to-cell communication that enables bacteria to sense the presence and number of other bacteria within their surrounding environment and to rapidly respond to changes in population density [134]. Autoinducers such as acyl-homoserine lactones (AHLs) are common signaling molecules for quorum sensing [135]. Zhang et al. [109] found that six AHLs could promote the conjugation frequency to varying degrees between the same bacteria genera during the advanced treatment of drinking water using biologically activated carbon.

#### 4.2.3. Bacterial Uptake of Extracellular ARGs

##### The Competent State of Bacteria

Bacterial cells must first develop a regulated physiological state of competence for natural transformation, which allows the occurrence of stable uptake, integration, and functional expression of extracellular DNA [53]. Taking *Bacillus subtilis* (*B. subtilis*) as an example, the formation of its competence requires the competency stimulating factor (CSF) [136], while the strong adsorption of CSF by kaolinite and montmorillonite reduced the transformation of ARGs [95]. In addition, the development of a competent state is also affected by various environmental stresses, such as population density, starvation, and DNA damage [137,138,139]. Soil bacterial communities normally live under conditions of starvation [140]. Inaoka et al. [137] noticed that the competent genes of *B. subtilis* 168 were up-regulated under these conditions and tended to be competent. Transformation is entirely directed by the recipient cell, and all required proteins are encoded in the core genome [141], so we should pay more attention to the gene expression of the recipient cell. Mineral-cell adhesion may influence the expression of competent genes in bacteria, thereby interfering with the development of a competent state [95].

##### Availability of Extracellular ARGs

As early as around 2000, the adsorption of DNA by soil components and the transformation activity of the adsorbed DNA have been thoroughly studied, not only for plasmids but also for chromosomes [99]. Some important components in soil can protect DNA from being degraded through the adsorption of nucleases so that it can be retained in the environment for a long time [12], and the adsorbed DNA still has transformation activity [98]. The interface that adsorbs DNA and the ion species or concentrations in the surrounding environment will affect the desorption and configuration of DNA [142]. Hu et al. [96] believed that the adsorption and desorption processes of ARGs by montmorillonite would cause a locally high concentration of ARGs around the montmorillonite particles, which was beneficial to the uptake of free ARGs by competent bacteria.

#### 4.2.4. Bacterial Concentration

The HGT process and the proliferation of ARB and ARGs in soil are intrinsically dependent on bacterial growth and concentration [111,112], especially for conjugation [143], but there are few studies on soil components that affect the concentration of bacteria and then influence the HGT of ARGs. The production of conjugants will be inhibited when the donor-to-recipient concentration ratio (R_D/R_) is too high [143,144,145]. Dahlberg et al. [144] found that the lowest concentration of donor bacteria created the highest conjugation frequency of plasmids.

## 5. Conclusions

Through the sorting and summary provided by this paper, it is found that ARB and ARGs have been widely detected in soil around the world, and the proliferation and spread of ARGs through HGT, VGT, and cooperative resistance are very popular. At present, scholars have explored the effects of soil types and soil components on the occurrence and transfer of ARGs, but related studies mainly focus on conjugation, and few systematic studies discuss the impact of multiple soil components. Meanwhile, the mechanisms such as cell membrane damage and the level of intracellular oxidation always gained more attention in the existing literature, ignoring the configurational changes of ARGs and the gene expression of the recipient during transformation under the action of soil components. In addition to genetic resistance, non-inherited resistance such as cooperative resistance has not received extensive attention. In order to further realize the occurrence and spread of ARGs in soil, Subsequent research needs to be carried out in the following aspects: (1) Breaking through the limitations of single-component research, mixed-component experiments, and multi-component systematic experiments should be used to further explore the effects of components in actual soil on ARGs. (2) Combining multi-omics methods such as genomics, proteomics, and transcriptomics to reveal the specific pathways and mechanisms of microbial responses under the effect of soil components. (3) Extending the research object from a single species to the intra-species and inter-species interactions about ARGs of different microorganisms in soil, comprehensively exploring the transfer and enrichment of ARGs between microbial communities under the treatment of different soil components.

## 6. Methods

We found 274 results in the Web of Science core collection database with the keywords “antibiotic resistance gene” and “soil component”, including 35 reviews. The research on soil components and ARGs has received extensive attention in recent years, and the number of related studies in 2019–2021 has increased significantly compared with previous years (Figure 2), indicating that this issue is gradually becoming a research hotspot in the field of the environment. The 35 reviews mainly focused on ARGs pollution, antibiotic removal and ARGs control, microbial regulation, and the correlation between heavy metals and ARGs in soil, while others were from agriculture and clinical medicine. However, there has been no review report on the effects of soil type and component on the HGT of ARGs and its mechanisms, lacking systematic analysis.

## Figures and Tables

**Figure 1 antibiotics-12-00333-f001:**
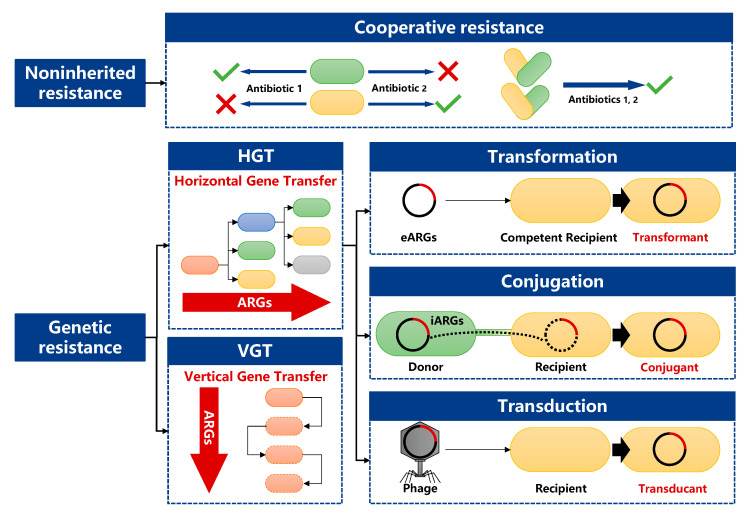
Transmission routes of antibiotic resistance genes (ARGs: antibiotic resistance genes, eARGs: extracellular ARGs, iARGs: intracellular ARGs).

**Figure 2 antibiotics-12-00333-f002:**
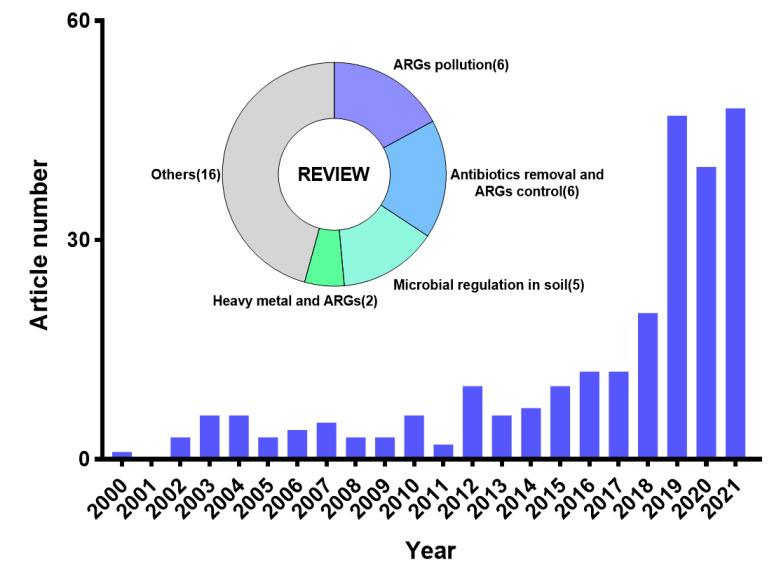
Search results and thematic analysis of the reviews.

**Table 1 antibiotics-12-00333-t001:** Antibiotic-resistance genes (ARGs) contamination in soil (copies/16S rRNA gene).

Place	Soil Type	ARGs	Relative Abundance
China [26]	Feedlot vicinity	*tet*M, *tet*O, *tet*Q, *tet*W	10^−5^–10^−2^
China [27]	Feedlot vicinity	*tet*B(P), *tet*M, *tet*O, *tet*W	10^−3^–10^−6^
China [28]	Feedlots	*tet*A(P), *tet*G, *tet*C, *tet*L, *tet*X, *tet*M, *tet*A	10^−2^–10^−4^
China [29]	Feedlots	*tet*A, *tet*B, *tet*M	10^−6^–10^−1^
China [30]	Farmland	*tet*B(P), *tet*M, *tet*O, *tet*Q, *tet*T, *tet*W	10^−8^–10^−2^
China [31]	Farmland	*tet*G, *tet*Y, *tet*Z	10^−7^–10^−4^
China [32]	Farmland	*tet*B(P), *tet*C, *tet*G, *tet*L, *tet*O, *tet*S, *tet*W, *tet*Z	10^−6^–10^−1^
Italy [33]	Feedlots	*tet*Q, *tet*W	10^−9^–10^−5^
India [34]	Feedlots	*tet*A, *tet*W	10^−1 a^
America [35]	Farmland	*tet*O, *tet*W	10^−7^–10^−4^
Austria [36]	Farmland	*tet*W	10^−5^–10^−4^
The Netherlands [24]	Typical sites	*tet*M, *tet*O, *tet*Q, *tet*W	10^−4^–10^−2^
Scotland [16]	Typical sites	*tet*M, *tet*Q, *tet*W	10^−5^–10^−2^
Scotland [37]	Farm	*tet*A, *tet*B, *tet*C, *tet*G, *tet*W	10^−6^–10^−5^
Australia [38]	Residential area	*tet*M, *tet*W	10^−9^–10^−2^

^a^ The ratio of phages carrying ARGs to the total number of phages.

**Table 4 antibiotics-12-00333-t004:** Research on the effects of soil components on transformation of antibiotic resistance genes (ARGs).

Medium	Important Conclusions	
	Results	Reasons
Kaolinite, illite, and montmorillonite [12]	Plasmids adsorbed on minerals could resist higher concentrations of nucleases and form more transformants than free plasmids.	The adsorption of the nuclease on minerals protected the plasmids, but it can still be involved in transformation.
Kaolinite, Goethite, and montmorillonite [95]	Low concentrations (1–2 g/L) have little effect; high concentration (10 g/L) of kaolinite and montmorillonite inhibited transformation; high concentration (10 g/L) of goethite promoted transformation.	Kaolinite and montmorillonite: strong adsorption to competence stimulating factor, decrease the expression level of competent genes (*phr*C, *com*S); goethite: increase cell membrane damage.
Montmorillonite [96]	Low concentration (about 0–0.025 g/L) promoted transformation; high concentration (about 0.025–2 g/L) inhibited transformation.	Low concentration: increase the contact between plasmids and cells; forming holes on cell membrane; High concentration: plasmids were adsorbed; heavy metals released from montmorillonite cause the aggregation of the plasmids.
Biochar [97]	Significantly inhibited the transformation of extracellular antibiotic resistance genes (eARGs)	Biochar dissolutions: Induce intramolecular condensation and agglomeration of plasmids; decrease the cell membrane permeability; biochar solids: Adsorb plasmids and deactivate *E. coli*.
Soil microcosm [98]	DNA adsorbed on soil particles still transformed competent cells	Minerals did not inhibit the transformation, but blocked DNA contact with the recipient.
Soil microcosm [99]	Plasmid adsorbed on sand transformed significantly less efficient than did plasmid in solution; the transformation by sand-adsorbed chromosomal was as high as that by plasmid in solution.	Transformation occurred by direct uptake of DNA from the mineral surfaces; transformation requires multiple plasmids, and the probability of multiple free plasmids meeting bacteria at the same time is higher than that on mineral surfaces; the chances of bacteria taking up DNA on the mineral surface are proportional to the size of the DNA, and chromosomes of the same mass are larger and easier to take up.
Activated sludge EPS [100]	The transformation ability of free ARGs was higher than that in activated sludge extracellular polymeric substances (EPS) when calculated per ng DNA, and lower when calculated per g volatile suspended solids.	Activated sludge EPS is rich in ARGs.
Sediment [101]	The transformation efficiency of adsorbed eARG was higher than that of free eARGs.	Sand adsorbed bacteria and plasmids at the same time, facilitating contact between the two, and was related to the conformation of the plasmid.

## Data Availability

Not applicable.

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
