# Peer review of "Soil Component: A Potential Factor Affecting the Occurrence and Spread of Antibiotic Resistance Genes"

_antibiotics, 2023, doi:10.3390/antibiotics12020333_

Round 1

Reviewer 1 Report

General comment: This work aimed to review the effects of soil type and soil components on the occurrence of antimicrobial resistance genes, and to analyse the mechanisms of soil components on genetic antimicrobial transfer.

This is an interesting review that includes a complete reference list with complete information about the issue.

Missing references in lines 29, 31, 86, 88, 107, 159, 215,231, 258, 265, 287.

Specific comments:

1.       Table 1:

1.1   Please verify and correct the relative abundance of the second line (China, reference 25)

1.2   Please verify and correct the antimicrobial resistance genes (ARGs) and relative abundance of the 3th line (China, reference 26). The relative abundance showed is not related to the AGRs mentioned.

1.3   Please include in the legend the abbreviations used in the Table.

2.       Line 136, verify the sentence.  You probably want to say “….high frequency.”

3.       Please improve the separation of each line on table 2, 3 and 4. It is difficult to read the text in each line without any barrier.

4.       In table 3 the last point does not have a described reason.

5.       What do you mean by pure bacterial system?

6.       Please include the name of the abbreviations in line 255 and 354.

7.       Which multi-omics methods do you suggest to use in the conclusion section? specify.

Reviewer 2 Report

The study, “Soil component: a potential factor affecting the occurrence and spread of antibiotic resistance genes” has added value to understand the mechanism of microbial genes in the soil. Overall study is interesting and important. However, some technical error during study have as follows: The design of the study is unclear. The sub-headings did not justify the content properly. Content was not properly organized.  

Some specific comments are as below:

Line 29-30: Important information please add citation

Line 43-52: This paragraph comes under method section, better to move in method section.

Design of the study is unclear. The sub-headings did not justify the content properly. Content was not properly organized.  

Line 78: Please clarify this sentence, “ARGs of tetracycline are one of the fastest growing keys

ARGs”.

Line 84-175: Transmission route can be summarized in tabulation

Reviewer 3 Report

Dear Authors, your review titled "Soil component: a potential factor affecting the occurrence and spread of antibiotic resistance genes" is very interesting as it focuses on the effects of soil type and components on HGT of antibiotic resistance genes that have never been investigated.

Your paper gives a clear panorama on how soil and its different components can enter the dynamics of AMR and suggests what are still the aspects to be implemented.

With regard to point 3 of your conclusions, are you sure that there is no data in the literature to begin with?

In Table 2 please control in the different column that  the first letter is in capital letter

Round 2

Reviewer 1 Report

The authors improved their manuscript.

The suggestions and indications mentioned in the first report were included in the new document.

Reviewer 2 Report

Authors have revised all the comments that I have assigned, the current version is okay.

Reviewer 3 Report

No other comments